# Plasticity of olfactory bulb inputs mediated by dendritic NMDA-spikes in rodent piriform cortex

**Amit Kumar[1], Edi Barkai[2], Jackie Schiller[1]\***

[1]Department of Physiology, Technion-Israel Institute of Technology, Haifa, Israel;
[2]Department of Neurobiology, University of Haifa, Haifa, Israel

**Abstract** The piriform cortex (PCx) is essential for learning of odor information. The current view postulates that odor learning in the PCx is mainly due to plasticity in intracortical (IC) synapses, while odor information from the olfactory bulb carried via the lateral olfactory tract (LOT) is 'hardwired.' Here, we revisit this notion by studying location- and pathway-dependent plasticity rules. We find that in contrast to the prevailing view, synaptic and optogenetically activated LOT synapses undergo strong and robust long-term potentiation (LTP) mediated by only a few local NMDA-spikes delivered at theta frequency, while global spike timing-dependent plasticity (STDP) protocols failed to induce LTP in these distal synapses. In contrast, IC synapses in apical and basal dendrites undergo plasticity with both NMDA-spikes and STDP protocols but to a smaller extent compared with LOT synapses. These results are consistent with a self-potentiating mechanism of odor information via NMDA-spikes that can form branch-specific memory traces of odors that can further associate with contextual IC information via STDP mechanisms to provide cognitive and emotional value to odors.

## Editor's evaluation

This article investigates the plastic properties of synapses impinging on pyramidal neurons in the piriform cortex from the lateral olfactory tract (LOT) and intracortical inputs. These findings uncover some of the location and pathway-dependent plasticity rules and challenge the notion that LOT inputs (carrying direct odor information from the bulb) become 'hardwired' after the critical period. The results provide novel information about how activity and experience alter synaptic communication in the olfactory circuit in a synapse-type specific manner.

**\*For correspondence:**
jackie@technion.ac.il

**Competing interest:** The authors declare that no competing interests exist.

## Introduction

The piriform cortex (PCx) is a main cortical station in olfactory processing, receiving direct odor information from the olfactory bulb (OB) via the lateral olfactory tract (LOT) as well as higher brain regions and is thought to be important for odor discrimination and recognition. Pyramidal neurons in the PCx serve as the main integration units within which the discrete molecular information channels are hypothesized to be recombined to form an 'odor object' (*Wilson and Sullivan, 2011*). PCx pyramidal neurons receive two spatially distinct inputs that terminate on different compartments of their apical dendrites: first, the direct afferent excitatory inputs from OB via the LOT, which terminate mainly on distal apical dendrites (layer Ia). Second, intracortical excitatory (IC) inputs from local PCx neurons and other cortical areas, which target the more proximal portions of the apical dendritic tree (layers Ib and II) and basal dendrites (*Bekkers and Suzuki, 2013*; *Haberly, 1985*; *Haberly, 2001*; *Haberly and Price, 1977*; *Isaacson, 2010*; *Neville and Haberly, 2004*; *Suzuki and Bekkers, 2006*).

Synaptic plasticity rules have crucial role in determining the way cortical networks acquire, organize, and store information. It is postulated that PCx mediates learning and recall of olfactory information (*Franks and Isaacson, 2005*; *Haberly, 1985*; *Haberly, 2001*; *Saar et al., 1998*; *Saar et al., 2001*). Previous studies have shown that similar to neocortical and hippocampal pyramidal neurons (*Bliss and Collingridge, 1993*; *Feldman, 2012*; *Sjöström et al., 2007*), IC synapses of PCx pyramidal neurons undergo plasticity changes. NMDA-R-dependent long-term potentiation (LTP) was robustly observed in IC synapses both when stimulated by theta rhythm alone (*Haberly et al., 1994*) or when paired with either a burst of LOT activation (*Kanter and Haberly, 1990*; *Kanter and Haberly, 1993*; *Kanter et al., 1996*), or back-propagating action potentials (BAPs) using spike timing-dependent plasticity (STDP) protocols (*Johenning et al., 2009*). In line with the importance of IC plasticity changes in learning, it was shown that following olfactory rule learning in vivo, PCx pyramidal neurons exhibited increase in IC synaptic strength and excitability (*Lebel et al., 2001*; *Saar and Barkai, 2003*; *Saar and Barkai, 2009*; *Saar et al., 2001*).

In contrast to IC inputs, plasticity of afferent LOT synapses is yet an unresolved issue (*Haberly et al., 1994*; *Jung et al., 1990*; *Roman et al., 1993*). NMDA-dependent LTP using theta burst stimulation of LOT inputs was less robust and weak. Typically, 10–15% potentiation was observed in afferent fibers after a single theta burst train, with saturation of 12–25% after multiple trains (*Franks and Isaacson, 2005*; *Jung et al., 1990*; *Kanter and Haberly, 1990*; *Poo and Isaacson, 2007*; *Roman et al., 1993*). STDP protocols failed altogether to induce LTP in LOT synapses because of the severe attenuation of BAPs to distal apical dendrites of PCx pyramidal neurons, where LOT inputs terminate (*Haberly, 2001*; *Neville and Haberly, 2004*). Currently, LOT synapses located in distal apical dendrites are considered for the most part to be 'hardwired' after the sensory critical period (*Bekkers and Suzuki, 2013*; *Franks and Isaacson, 2005*), and robust NMDA-R-dependent LTP is limited to a brief postnatal time window of approximately 4 weeks (*Franks and Isaacson, 2005*; *Poo and Isaacson, 2007*), which is in contrast to IC synapses that retain the capacity for plasticity throughout adulthood. Yet, the notion of LOT synapse stability in adulthood is in conflict with reports obtained in vivo during complex olfactory learning (*Cohen et al., 2008*; *Cohen et al., 2015*; *Patneau and Stripling, 1992*).

We have recently reported the generation of local NMDA-spikes in distal apical dendrites by activation of LOT synapses (*Kumar et al., 2018*). These dendritic NMDA-spikes generate large localized calcium transients, which can serve as postsynaptic signals for LTP induction in the activated LOT synapses. In support of this hypothesis, it was recently shown that NMDA-spikes were critical for inducing LTP in the dendrites of CA3 pyramidal neurons (*Brandalise et al., 2016*) and in layer 2–3 pyramidal neurons of somatosensory cortex in vivo (*Gambino et al., 2014*; *Gordon et al., 2006*). We thus hypothesized that synchronized firing of spatially clustered afferent LOT synapses can generate local NMDA-spikes, which in turn can trigger LTP of the activated synapses in distal dendrites of PCx pyramidal neurons.

Consistent with this hypothesis and settling the inconsistencies in the literature, we found that local NMDA-spikes generated in distal apical dendrites delivered at 4 Hz mimicking the exploratory sniff cycle in rats (*Wilson, 1998*) can induce robust and strong LTP (>200%) of LOT inputs impinging on distal apical dendrites of PCx pyramidal neurons. Only a few NMDA-spikes (as low as two NMDA-spikes) were required for potentiation of LOT inputs. As previously described, STDP protocol failed to induce LTP of LOT synapses (*Johenning et al., 2009*). In contrast, for IC apical and basal synapses, potentiation was mediated by both NMDA-spikes and STDP protocols, but the magnitude of potentiation was smaller compared to that of LOT synapses.

## Results

The inputs to PCx are highly segregated along the axis of apical dendrites in principal pyramidal neurons. This raises the possibility of location specific plasticity rules as was previously shown for neocortical pyramidal dendrites (*Gordon et al., 2006*; *Kampa et al., 2007*; *Letzkus et al., 2006*; *Letzkus et al., 2007*; *Lisman and Spruston, 2010*; *Sandler et al., 2016*; *Sjöström and Häusser, 2006*; *Sjöström et al., 2008*). In this work, we tested both global STDP protocols as well as local induction protocols mediated by dendritic NMDA-spikes (*Kumar et al., 2018*).

## Location-dependent STDP in apical and basal dendrites of PCx pyramidal neurons

To directly test the location-dependent capability of PCx dendrites to undergo LTP with an STDP protocol, we performed whole-cell patch-clamp voltage recordings from the soma of layer II pyramidal neurons as determined from the Dodt contrast image and somatic firing pattern (*Suzuki and Bekkers, 2006*). Neurons were loaded with calcium-sensitive dye OGB-1 (200 μM) and CF633 (200 μM) to enable visualization of dendrites using a confocal microscope. We used focal synaptic stimulation to activate synaptic inputs in distal LOT (299 ± 6.29 μm from soma) synapses located in layer Ia (*Figure 1A*) or more proximally (139.33 ± 5.15 μm from soma) in layer Ib (*Figure 1E*) to activate predominantly the IC synapses (*Bekkers and Suzuki, 2013*; *Isaacson, 2010*; *Kumar et al., 2018*; *Suzuki and Bekkers, 2011*). LTP was induced by pairing a single excitatory post synaptic potential (EPSP; 1.88 ± 0.23 mV) with a burst of postsynaptic BAPs at 20 Hz (a sequence of three BAPs @ 150 Hz repeated three times at 20 Hz; each sequence was repeated 40 times at 5 s interval) that simulates the beta oscillation frequency of odor-evoked synaptic activity in PCx (*Johenning et al., 2009*; *Poo and Isaacson, 2007*; *Figure 1B*). In accordance with a previous report (*Johenning et al., 2009*), this paradigm failed to induce LTP in distally activated apical synapses at layer Ia where most of the LOT inputs terminate (*Figure 1C and D*; the EPSP amplitude after the induction protocol was 103.06% ± 2.81% of the control EPSP; p=0.7658; n = 11). Next, we replaced APs evoked by somatic current injection with postsynaptic APs evoked by synaptic stimulation in layer IIa (*Figure 1—figure supplement 1*). When paired with an EPSP (as described in *Figure 1B*), this STDP protocol also failed to potentiate distally located layer Ia inputs (104.99% ± 3.86% of control; p=0.7577, n = 3). In sharp contrast, using the same protocol but activating IC instead of LOT synapses at more proximal apical dendritic locations induced robust potentiation of layer Ib inputs (*Figure 1F and G*; 168.93% ± 6.54% of control EPSP; p=0.01387, n = 6).

PCx pyramidal neurons also possess an elaborated basal dendritic arbor (*Suzuki and Bekkers, 2006*). These dendrites receive IC inputs, mainly feedback inputs from local neurons in PCx and other olfactory areas (*Hagiwara et al., 2012*; *Isaacson, 2010*). We focally stimulated inputs innervating basal dendrites (129.25 ± 8.83 μm from the soma, n = 8) using the same STDP protocol as for apical dendrites (*Figure 2A*). We observed a significant LTP in these basally located synapses, however, with a relatively smaller magnitude (*Figure 2B–D*; 137.12% ± 6.11% of control EPSP; p=0.0031, n = 8) compared to potentiation values in proximal apical dendrites (p=0.00427) using the same induction protocol.

Thus, the STDP burst protocol is efficient in inducing robust LTP in IC inputs both at proximal apical locations and basal synapses but fails altogether to induce LTP at LOT synapses located on distal apical dendrites.

## Local NMDA-spikes induce LTP in distal LOT and proximal IC inputs of PCx pyramidal neurons

We recently reported the initiation of local NMDA-spikes in distal apical dendrites of PCx pyramidal neurons by activation of LOT synapses (*Kumar et al., 2018*). Dendritic spikes are excellent candidates for serving as a local postsynaptic signal for induction of long-term synaptic plasticity (*Gambino et al., 2014*; *Golding et al., 2002*; *Gordon et al., 2006*; *Lisman and Spruston, 2005*; *Lisman and Spruston, 2010*; *Major et al., 2013*; *Polsky et al., 2009*; *Remy and Spruston, 2007*).

To investigate the possible role of NMDA-spikes in inducing LTP of LOT inputs, we recorded from layer II pyramidal neurons and activated LOT synapses both synaptically and optogenetically. For synaptic stimulation, we focally stimulated LOT synapses impinging on distal layer Ia dendrites (272.75 ± 6.6 μm from the soma) and evoked local NMDA-spikes using a short burst (three-stimuli) at 50 Hz (*Figure 3A and B*). The LTP induction protocol consisted of 2–7 local NMDA-spikes repeated at 4 Hz, mimicking the exploratory sniff cycle (*Figure 3C*). This local induction protocol was very efficient in inducing robust and strong LTP of EPSPs in the distal LOT synapses terminating in layer Ia (213.98% ± 10.81%; p=0.0000687; n = 26; *Figure 3D and F*). As few as two full-blown NMDA-spikes at 4 Hz were enough to cause potentiation (*Figure 3E*), which lasted for the length of the recording (>75 min).

We next addressed the question whether NMDA-spikes were critical for potentiation of LOT inputs or whether EPSP subthreshold to NMDA-spike (sub-NMDA), delivered at 4 Hz, can also induce LTP of LOT inputs. EPSP subthreshold to NMDA-spike initiation delivered with same induction protocol

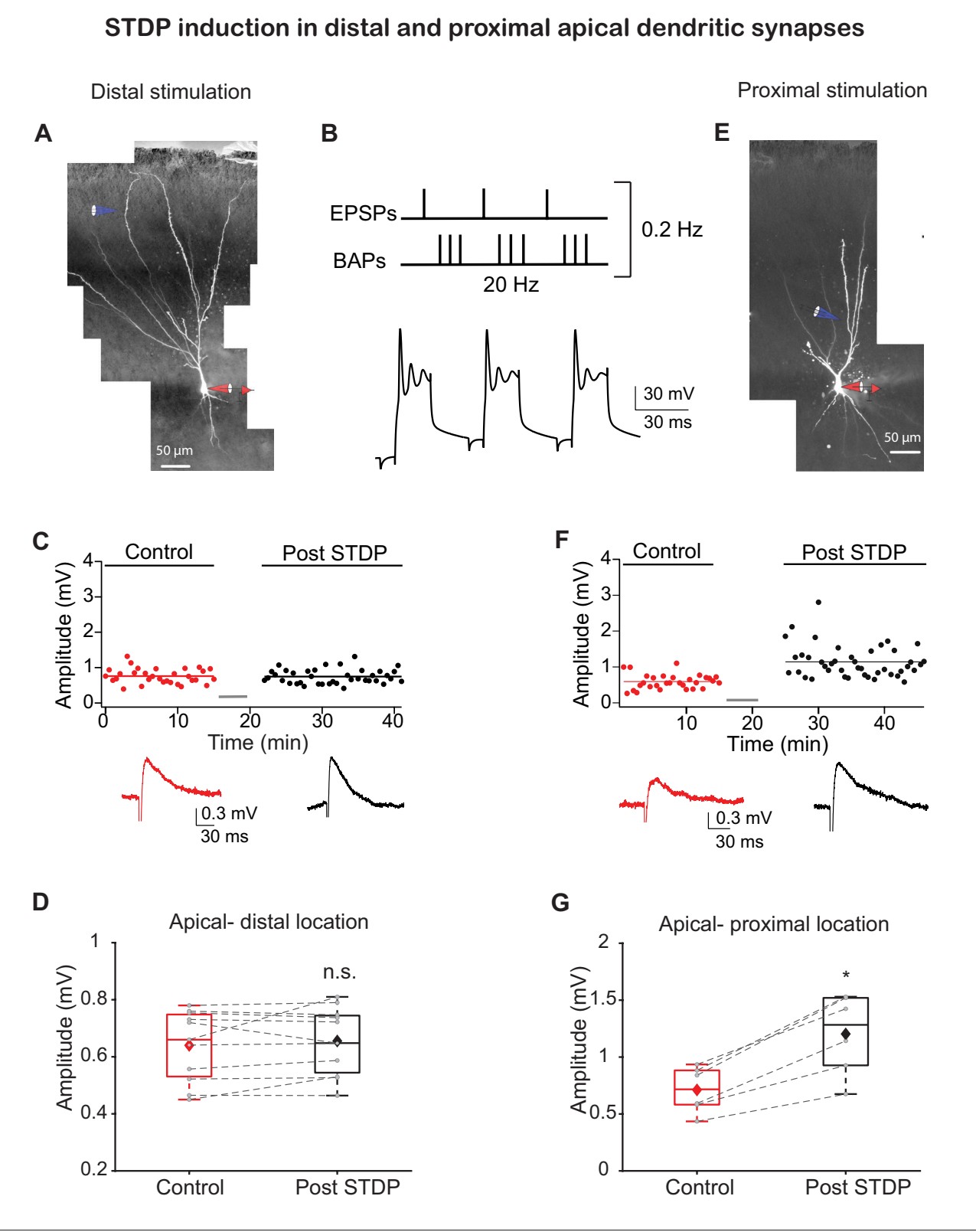

**Figure 1.** Spike timing-dependent plasticity (STDP) protocol induces long-term potentiation (LTP) in synapses at proximal apical but not distal dendritic locations. (**A**) Experimental setup. A pyramidal neuron from layer IIb was loaded with fluorescent dyes CF633 (200 µM) and OGB-1 (200 µM) via a somatic patch electrode (red electrode). Focal stimulation was performed using a theta electrode located nearby a distal apical dendrite in layer Ia (blue electrode; 299 ± 6.29 µm from soma), while recording at the soma. (**B**) STDP induction protocol (top): one EPSP followed (8 ms delay) by a burst of

*Figure 1 continued on next page*

*Figure 1 continued*

three back-propagating action potentials (BAPs) at 150 Hz, repeated three times at 20 Hz. Further, this triplet paired EPSP-BAPs was repeated 30 times at 0.2 Hz. Bottom: example of a voltage response to this induction pairing stimulus. (**C**) Amplitude of single EPSPs is represented over time for control stimulation and after STDP induction protocol at distal apical dendrite (gray bar represents the time of induction stimulus). Control EPSPs were recorded at 0.033 Hz for 10–15 min, prior and after STDP induction protocol. Bottom: traces of average EPSPs for control (red) and post induction (black) from the cell shown in (**A**). (**D**) Box plot showing EPSP amplitudes pre- and post-STDP induction for layer Ia inputs. No significant change was observed, post-STDP EPSP amplitude was 103.06% ± 2.81% of the control (p=0.7658; n = 11). (**E**) Same as in (**A**) for activation of proximal apical dendrite activation site (blue electrode; 139.33 ± 5.15 μm from soma) at layer Ib. (**F**) Same as in (**A**) for proximal apical dendritic activation site. Control EPSPs were recorded at 0.03 Hz for 10–15 min, prior and after STDP induction protocol. Bottom: traces of average EPSPs for control (red) and post-induction (black) from the cell shown in (**E**). (**G**) Box plot showing EPSP amplitudes pre- and post-STDP induction for layer Ib inputs. Layer Ib EPSPs were significantly enhanced post-induction protocol, 168.93% ± 6.54% of control (p=0.01387, n = 6). In box plots, the gray dots represent the average EPSP of each experiment, and the diamond represents the mean of the entire set. Dotted gray lines connect between pairs of control and post-induction values. Data for (**C**), (**D**), (**F**), and (**G**) can be found in *Figure 1—source data 1*. See also *Figure 1—figure supplement 1*.

The online version of this article includes the following figure supplement(s) for figure 1:

**Source data 1.** Data for *Figure 1C, D, F and G*.

**Figure supplement 1.** Pairing synaptically induced EPSPs and back-propagating action potentials (BAPs) also failed to induce long-term potentiation (LTP) of lateral olfactory tract (LOT) inputs.

**Figure supplement 1—source data 1.** Data for *Figure 1—figure supplement 1C and D*.

failed to evoke potentiation of the LOT inputs (*Figure 3H* and *Figure 3—figure supplement 1A and B*). The average EPSP amplitude after sub-NMDA-spike induction was 106.79% ± 5.69% of the control EPSP (p=0.8373, n = 5 cells). Also, in experiments where NMDA-spikes were blocked by application of the NMDA-R antagonist DL-2-amino-5-phosphonovaleric acid (APV; 50 μM), LOT synapses failed to undergo potentiation using the same induction protocol (*Figure 3F* and *Figure 3—figure supplement 1C and D*). The EPSP amplitude post induction was 95.06% ± 4.69% of control EPSP in the presence of APV (p=0.817, n = 5).

To ensure that the distal NMDA-spikes and local EPSPs were indeed evoked primarily by stimulation of LOT inputs, we performed two additional experiments: the first was using the GABA_B agonist baclofen (100 μM), which has been shown to preferentially silence IC inputs (*Apicella et al., 2010*; *Franks and Isaacson, 2005*; *Kumar et al., 2018*; *Tang and Hasselmo, 1994*). Application of baclofen did not change the resting membrane potential and did not alter the chances of occurrence of NMDA-spikes (*Kumar et al., 2018*). Bursts of NMDA-spikes delivered at 4 Hz in the presence of baclofen also caused potentiation of layer Ia inputs (*Figure 4A–D*; 236.57% ± 17.51% of the control EPSP recorded at 0.033 Hz; p=0.0246; n = 5 cells). In the second set of experiments, we activated LOT inputs optogenetically using ChR2 viral transfection to the bulb (*Figure 5*). In these experiments, EPSPs were generated by optogenetic activation of LOT axons nearby a distal apical dendrite (~5 μm² illumination spot) and NMDA-spikes were generated by either more intense optogenetic activation of LOT inputs or glutamate uncaging (MNI-Glutamate) (*Figure 5A and B*). Only three pairings of opto-EPSPs and NMDA-spikes were sufficient in inducing robust and strong LTP of the LOT inputs (232.45% ± 16.55%; p=0.000286; n = 11; *Figure 5A–D*).

A similar NMDA-spike induction protocol (4–10 NMDA-spikes at 4 Hz) also induced potentiation of proximal apical IC synapses (127 ± 4.39 μm from soma) (*Figure 6A–D*), but to a smaller extent compared to distal apical LOT synapses and even smaller than potentiation induced by STDP protocol in these IC synapses. Post induction, the apical IC EPSP amplitude was 148.48% ± 4.1% of the control (*Figure 6F*; p=0.00204; n = 9; p=0.0013 for comparison of proximal versus distal NMDA-spike potentiation; p=0.0127 for comparison with STDP in IC synapses). Typically, full-blown NMDA-spikes initiated at these proximal apical locations were accompanied with BAPs (*Figure 6B*). To control for the contribution of these BAPs, we repeated the induction protocol, but instead of using NMDA-spikes we used pairing of BAPs and local EPSPs for five repetitions (one EPSP paired with three BAPs at 150 Hz repeated five times at 4 Hz). In this case, we did not observe potentiation of these proximal IC synapses (*Figure 6G*), thus we concluded that NMDA-spikes were crucial for the potentiation of IC synapses with the NMDA-spike protocol. This EPSP + BAPs pairing paradigm differed from the STDP protocol used in *Figure 1* in terms of the number of repetitions, frequency, and degree of potentiation (p<0.0003 for comparing the potentiation between the two pairing paradigms). Interestingly, calcium transients evoked by NMDA-spikes both in spines and adjacent shafts were significantly larger

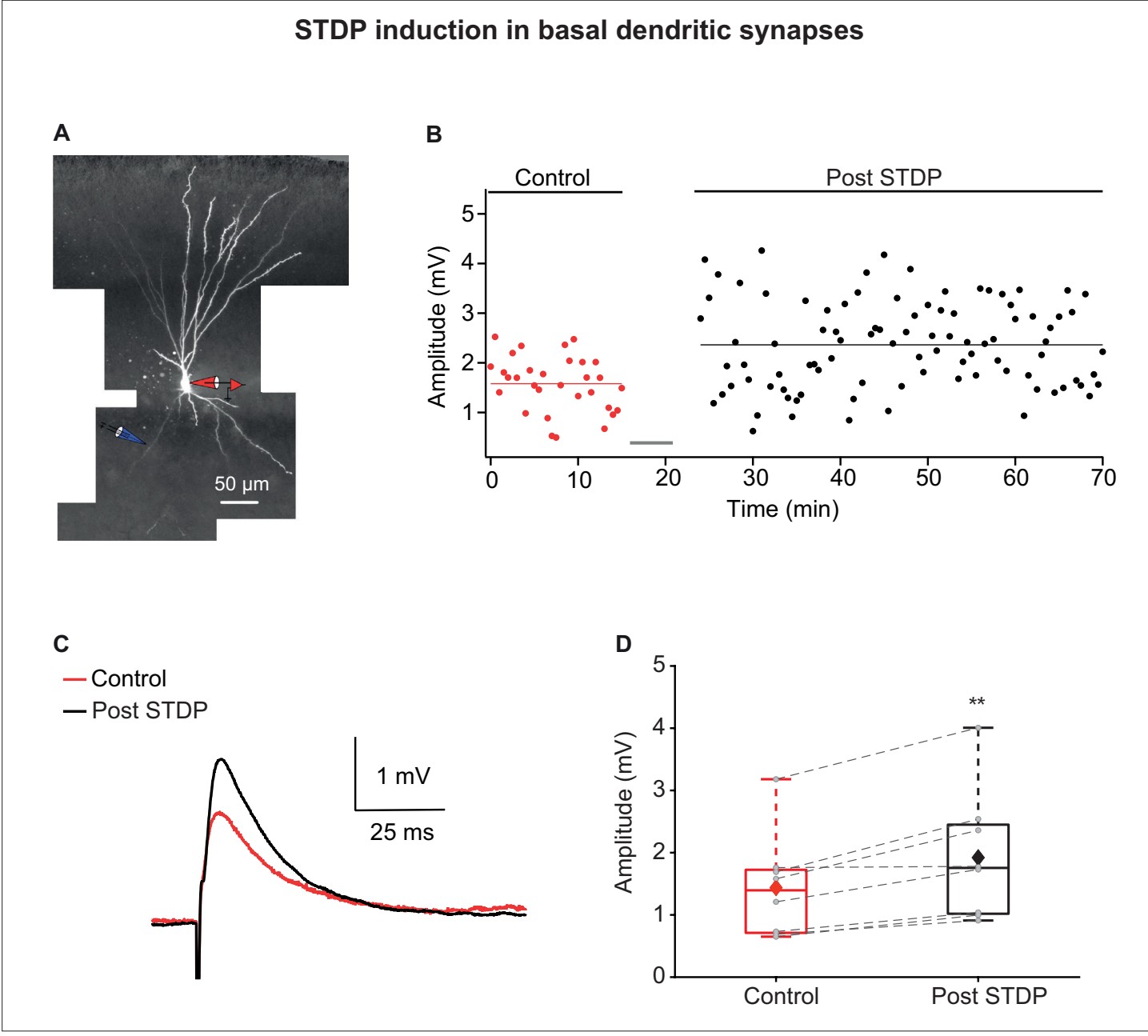

**Figure 2.** Long-term potentiation (LTP) induction via spike timing-dependent plasticity (STDP) protocol in basal dendrites of piriform cortex (PCx) pyramidal neurons. (**A**) A layer IIa pyramidal neuron filled with CF633 (200 µM) and OGB-1 (200 µM) via the somatic recording electrode (red electrode). Focal stimulation was performed using a double-barrel theta electrode located nearby a distal basal dendritic site (blue electrode; 129.25 ± 8.83 µm from soma). (**B**) Amplitude of single EPSPs is represented over time for control stimulation and after STDP induction protocol (gray bar represents the time of induction stimulus). Induction protocol, as described in *Figure 1B*. Bottom: traces of average EPSPs from the cell shown in (**A**) for control (red) and post induction (black). (**C**) Traces of average EPSPs evoked during control (red) and after STDP induction (black). (**D**) Box plot showing EPSP amplitudes pre- and post-STDP induction during control (red), and post-STDP induction (black) for distal basal input stimulation. EPSPs were significantly enhanced post-induction protocol (137.12% ± 6.11% of the control; p=0.0031, n = 8). The gray dots represent the average EPSP for each cell, and the diamond represents the mean EPSP of the entire set. Dotted gray lines connect between pairs of control and post-induction values.

The online version of this article includes the following figure supplement(s) for figure 2:

**Source data 1.** Data for *Figure 2B and D*.

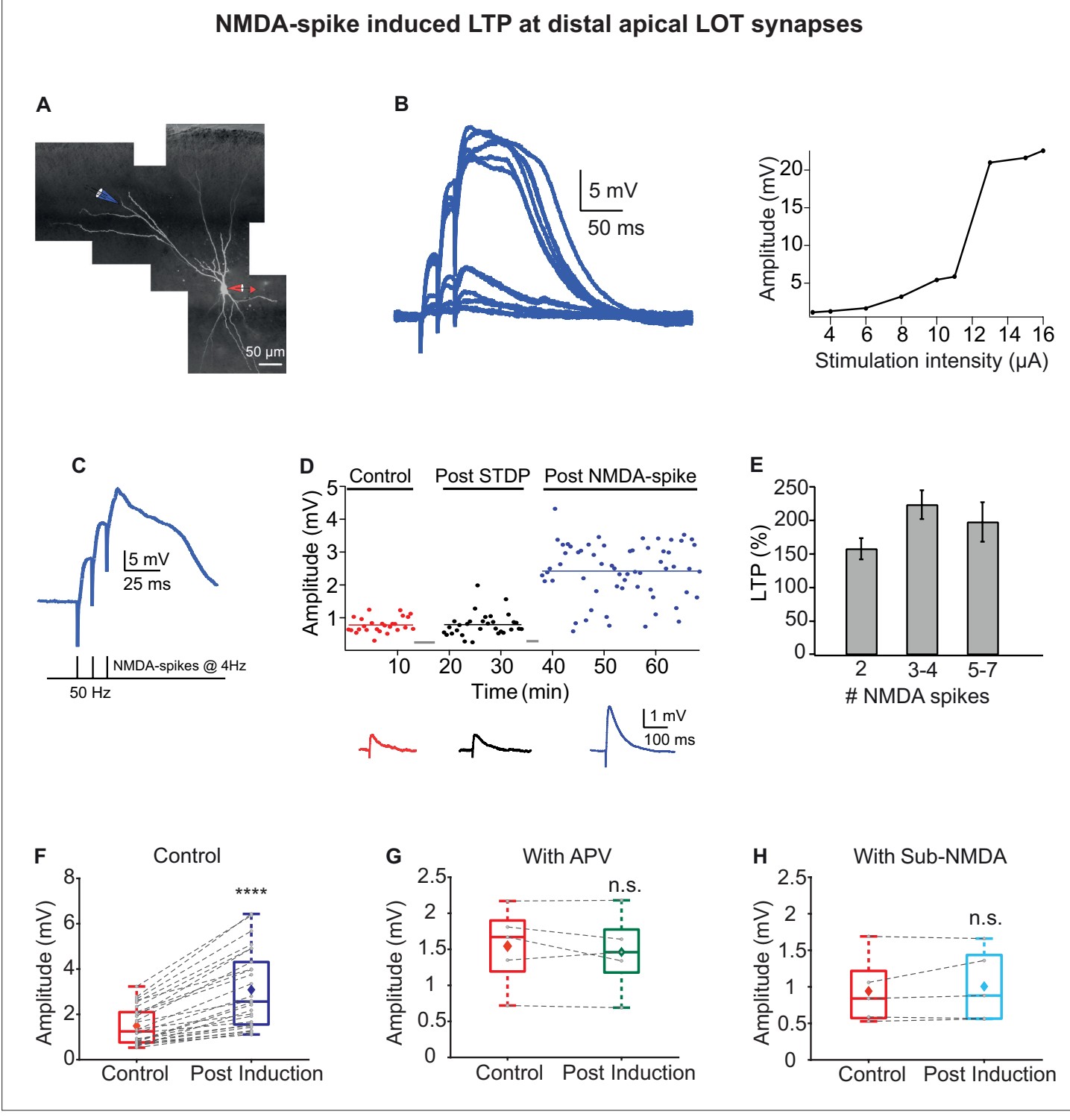

**Figure 3.** Long-term potentiation (LTP) of lateral olfactory tract (LOT) inputs by NMDA-spikes in piriform cortex (PCx) pyramidal neurons. (**A**) Fluorescence image of a layer IIa pyramidal neuron filled with CF633 (200 µM) and OGB-1 (200 µM) via the patch recording electrode (red electrode). A focal synaptic stimulating theta electrode was placed at the distal apical dendrite at layer Ia (blue electrode; 272.75 ± 6.6 µm from the soma). (**B**) Voltage responses evoked by gradually increasing synaptic stimulation intensity (burst of three pulses at 50 Hz). Peak voltage response as a function of stimulus intensity showing an all-or-none response (left). (**C**) Schematic of NMDA-spike induction protocol (bottom). NMDA-spikes evoked by three pulses at 50 Hz, repeated at 4 Hz for 2–7 times. Upper panel: example voltage response to NMDA-spikes induction protocol. (**D**) Amplitude of single EPSPs is represented over time for control stimulation (red), after STDP induction protocol (black) and after NMDA-spike induction protocol (blue). Gray bars represent the time of induction stimulus. Control EPSPs were recorded at 0.033 Hz for 10–15 min. Potentiation was observed only after the

*Figure 3 continued on next page*

*Figure 3 continued*

NMDA-spike induction protocol. Bottom: traces of average EPSPs in control (red), post-STDP (black), and post-NMDA-spike induction (blue) for the cell shown in (**A**). (**E**) Plot of % LTP (relative to control EPSPs) vs. number of NMDA-spikes evoked during the induction protocol. All values are insignificant. (**F**) Box plot showing the EPSP amplitude during control (red) and post-NMDA-spike induction protocol (blue). NMDA-spike induction protocol induced large potentiation of the control EPSP (213.98% ± 10.81%; p=0.0000687; n = 26). (**G**) Box plot showing the EPSP amplitude during control NMDA-spike induction protocol (red) and after induction in the presence of APV (50 µM; green). No significant change in EPSP was observed when NMDA-spikes were blocked with APV (95.06% ± 4.69% of control; p=0.817, n = 5). (**H**) Box plot showing the EPSP amplitude for control NMDA-spike induction protocol (red) and after induction with sub-NMDA EPSPs (teal). No significant change in EPSP amplitudes was observed (106.79% ± 5.69%, p=0.8373, n = 5). In box plots, the gray dots represent the average EPSP for each cell, and the diamond represents the mean EPSP of the entire set. Dotted gray lines connect between pairs of control and post-induction values. See also *Figure 3—figure supplements 1 and 2*.

The online version of this article includes the following source code and figure supplement(s) for figure 3:

**Source data 1.** Data for *Figure 3B, D, F-H*.

**Figure supplement 1.** Example experiments showing NMDA-spikes are necessary for long-term potentiation (LTP) of lateral olfactory tract (LOT) synapses.

**Figure supplement 1—source data 1.** Data for *Figure 3—figure supplement 1B and D*.

**Figure supplement 2.** Estimation of the number of synapses needed to initiate NMDA-spikes with in vivo-like odor stimulation.

**Figure supplement 2—source code 1.** Source code for *Figure 3—figure supplement 2* is provided.

than those evoked by STDP stimulation (*Figure 6E and H*; p<0.0001), yet the degree of potentiation with STDP protocol was higher (p=0.0127) compared to the NMDA-spike protocol. This result indicates that the amount of calcium entry per se is not the only variable determining the degree of potentiation.

Taken together, these results confirmed that NMDA-spikes are a strong mediator for potentiating both distal LOT and proximal IC inputs (see *Figure 3—figure supplement 2* and Discussion regarding the feasibility for initiating NMDA-spikes with odor stimulation under in vivo conditions).

## Local NMDA-spikes can be initiated and induce potentiation in basal dendrites of PCx pyramidal neurons

Basal dendrites in PCx pyramidal neurons are also a major target for IC connections (*Haberly, 1985*; *Isaacson, 2010*; *Luskin and Price, 1983a*; *Luskin and Price, 1983b*). Presently, it is unknown whether basal dendrites can generate local NMDA-spikes and in turn whether these spikes can serve to induce plasticity in these dendritic locations.

To test this possibility, we focally stimulated distal basal dendrites using visually positioned theta electrode placed at single basal dendrites with an average distance of 144.83 ± 9.21 µm from the soma (*Figure 7A*). Similar to apical dendrites, dendritic NMDA-spikes were evoked in basal dendrites of PCx neurons (*Figure 7B*). The average spike threshold recorded at the soma was 10.2 ± 1.6 mV. The dendritic spike amplitude and area as measured at the soma was 27.2 ± 2.5 mV and 2999.5 ± 434.7 mV * ms, respectively (n = 6 cells).

Next, we tested if these NMDA-spikes can mediate potentiation of basal dendrite inputs. Using the same NMDA-spike induction protocol used for apical dendrites significantly potentiated the IC synapses on basal dendrites (*Figure 7C*; 140.64% ± 4.5% of the control; p=0.006526, n = 6 cells). However, the degree of potentiation was smaller compared to that of LOT inputs in distal apical dendrites (*Figure 7D*). EPSPs that were subthreshold for NMDA-spikes failed to induce significant potentiation of the EPSPs (108.73% ± 3.13% of the control EPSP, p=0.8253, n = 5; *Figure 7F* and *Figure 7—figure supplement 1C and D*). The same induction protocol in the presence of NMDA-R blocker APV (50 µM) failed to cause potentiation at these basal synapses (100.69% ± 4.27% of control, p=0.9948, n = 5; *Figure 7E* and *Figure 7—figure supplement 1A and B*). These results indicate that local NMDA-spikes are necessary for inducing potentiation of basal inputs.

## Discussion

The PCx was shown to be critical for odor discrimination, recognition, and memory, and thus, odor memory was attributed to plasticity changes in the PCx (*Ghosh et al., 2016*; *Haberly, 2001*; *Hasselmo and Barkai, 1995*; *Saar et al., 2012*). However, it is still unclear which synapses and pathways are the

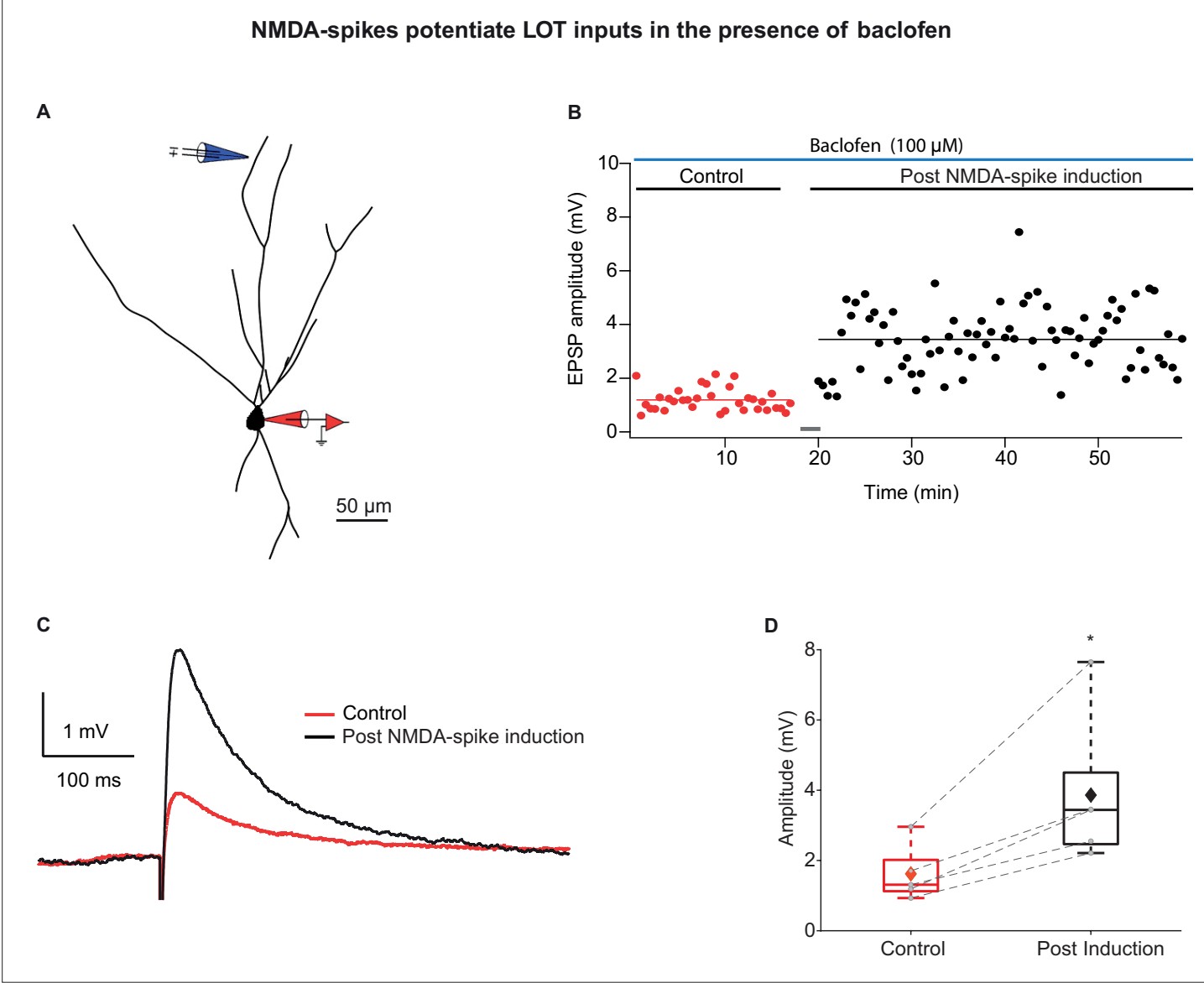

**Figure 4.** Long-term potentiation (LTP) of lateral olfactory tract (LOT) inputs by NMDA-spikes while blocking intracortical excitatory (IC) inputs with baclofen. (**A**) Reconstruction of layer IIa pyramidal neuron filled with CF633 (200 µM) and OGB-1 (200 µM) via the patch recording electrode (red). A focal double-barreled synaptic stimulating electrode was placed at the distal apical dendrite in layer Ia (blue; 287 ± 4.34 µm from soma). (**B**) Amplitude of single EPSPs represented over time for control stimulation and after NMDA-spike induction protocol at distal apical dendrite in the presence of GABA$_B$ agonist baclofen (100 µM). Gray bar represents the time of induction stimulus. Control EPSPs were recorded at 0.033 Hz (red), prior and post-induction protocol (black). (**C**) Average amplitude of EPSPs during control (red) and post-NMDA-spike induction protocol (black). (**D**) Box plot showing EPSP amplitudes pre- (red) and post-LTP induction (black) for layer Ia inputs. Layer Ia EPSPs were significantly enhanced post-induction protocol in the presence of baclofen, 236.57% ± 17.51% of control EPSP (p=0.0246, n = 5). The gray dots represent the average EPSPs of each experiment, and the diamond represents mean of the entire set. Dotted gray lines connect between pairs of control and post-induction values.

The online version of this article includes the following figure supplement(s) for figure 4:

**Source data 1.** Data for **Figure 4B and D**.

target of these plasticity changes. Here, we challenge the notion that LOT inputs become 'hardwired' after the olfactory critical period (*Franks and Isaacson, 2005*; *Kanter and Haberly, 1993*; *Poo and Isaacson, 2007*), and only IC inputs undergo plasticity changes in adulthood. We show that while LOT inputs contacting distal apical PCx pyramidal neuron dendrites do not undergo significant LTP when paired with somatic output activation using STDP plasticity protocols, they undergo strong and robust LTP when co-activated with other spatially clustered LOT inputs that generate local NMDA-spikes.

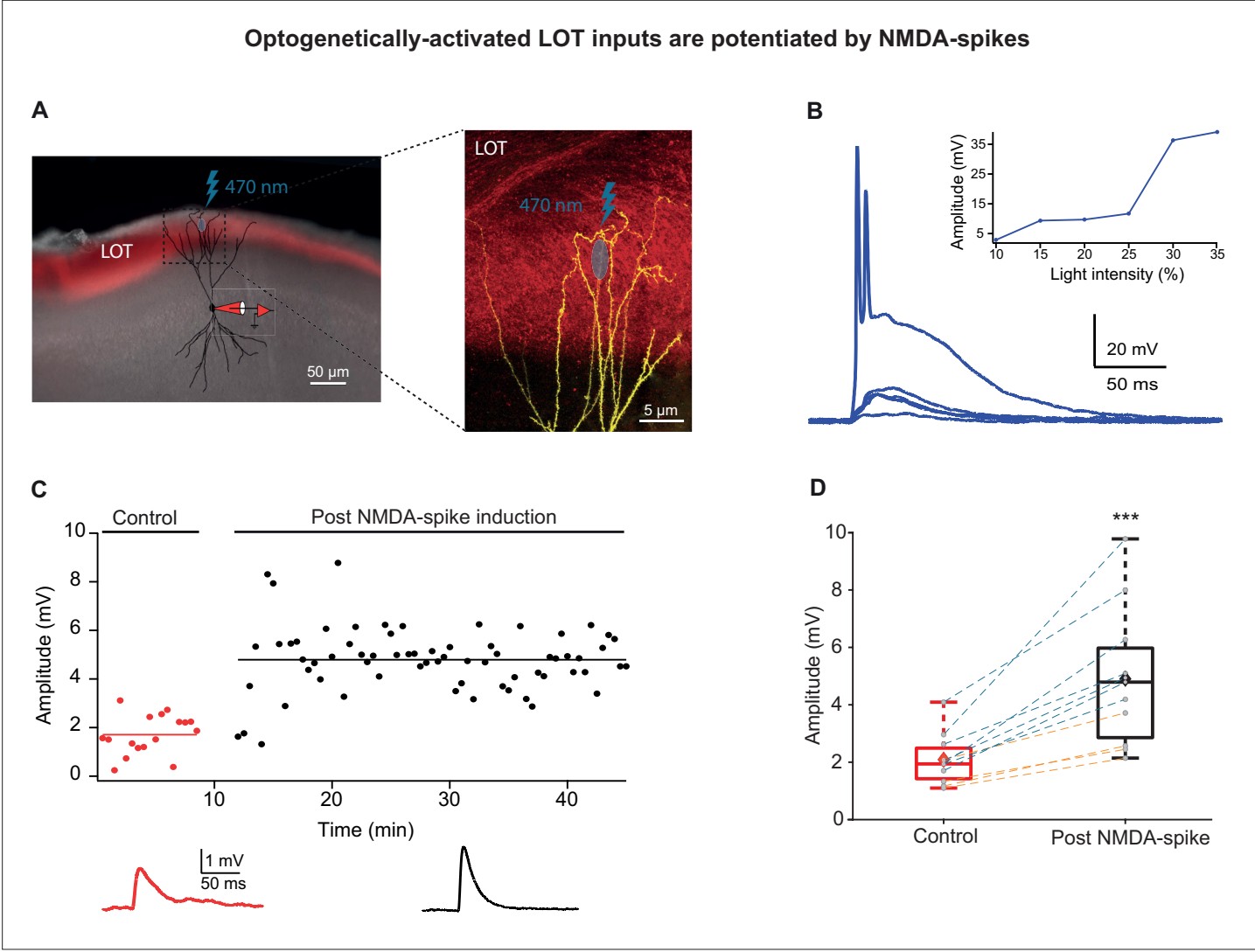

**Optogenetically-activated LOT inputs are potentiated by NMDA-spikes**

**Figure 5.** Optogenetically activated lateral olfactory tract (LOT) inputs were potentiated by NMDA-spikes. (**A**) Experimental setup. Left panel: coronal slice of piriform cortex (PCx) from a mouse previously injected in olfactory bulb (OB), with a virus expressing ChR2 (pAAV.CAG.hChR2(H134R)-mCherry. WPRE.SV40) in LOT fibers (red fluorescence). A pyramidal neuron from layer IIb was loaded with CF633 (200 μM) and OGB-1 (200 μM) via a somatic patch electrode (red electrode) and was reconstructed after the recording session. Right panel: opto-EPSPs and NMDA-spikes were evoked by light stimulation (LED 470 nm) directed to a small portion of the distal apical dendrite (~5 μm²). (**B**) Voltage responses evoked by gradually increasing ontogenetic light intensity (three pulses of 5 ms at 50 Hz). Peak voltage responses as a function of % light intensity showing an all-or-none spike response (inset). (**C**) Amplitude of single opto-EPSPs is represented over time for control stimulation and after NMDA-spike induction protocol at distal apical dendrite. Bottom panel: average opto-EPSP in control (red), post-NMDA-spike induction (black). (**D**) Box plot showing opto-EPSP amplitudes pre- and post-NMDA-spike induction for LOT inputs. Opto-EPSPs were significantly enhanced post-NMDA-spike optogenetic induction protocol, 232.45% ± 16.55% of control (p=0.000286; n = 11). The gray dots represent the average EPSP of each experiment, and the diamond represents the mean of the entire set. Experiments with optogenetic activation of LOT inputs (blue lines) and experiments with glutamate uncaging (orange lines) were pooled together. There was no significant difference in the level of potentiation between these two activation methods (194.2% ± 9.2%, n = 4, and 254.3% ± 23.6%, n = 7, for glutamate uncaging and opto-EPSPs, respectively, p=0.08). Dotted gray lines connect between pairs of control and post-induction values.

The online version of this article includes the following figure supplement(s) for figure 5:

**Source data 1.** Data for *Figure 5B-D*.

Only a few (≥2) NMDA-spikes are necessary for full-blown LTP induction in these distal LOT synapses. In contrast to distal LOT inputs, IC inputs in more proximal apical and basal dendritic locations can undergo plasticity changes by both global STDP protocol and local NMDA-spikes, albeit to a smaller extent compared with LOT synapses.

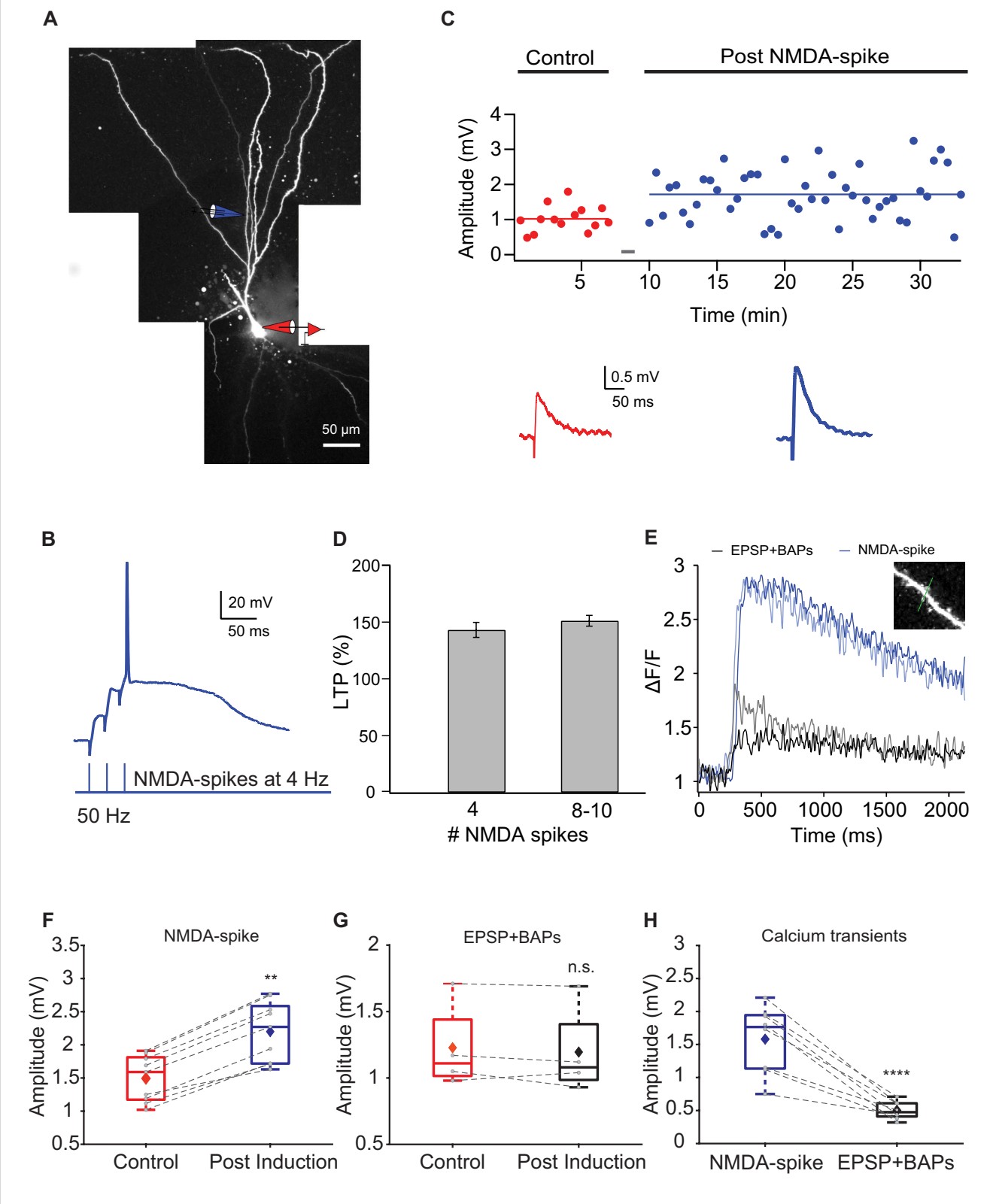

**Figure 6.** Long-term potentiation (LTP) of layer Ib intracortical (IC) inputs evoked by local NMDA-spikes. (**A**) Fluorescence reconstruction of a layer IIa piriform cortex (PCx) pyramidal neuron filled with CF633 (200 µM) and OGB-1 (200 µM) via the somatic patch electrode (in red). Focal stimulation was performed using a double-barrel theta electrode located nearby a proximal apical site in IC layer Ib (in blue; 127 ± 4.39 µm from soma). (**B**) Schematic of NMDA-spike induction protocol: NMDA-spikes evoked by three pulses at 50 Hz, repeated at 4 Hz for 4–10 times. Upper panel: an example trace of

*Figure 6 continued on next page*

*Figure 6 continued*

NMDA-spike delivered at 4 Hz during the LTP induction protocol. (**C**) Amplitude of single EPSPs represented over time during control (red) and post-NMDA-spike induction (blue). Gray bars indicate the time of the LTP induction protocol. Lower panel: average EPSP in control (red) and post-NMDA-spike induction (blue). (**D**) Plot of % LTP (relative to control EPSPs) vs. number of NMDA-spikes evoked during the induction protocol. All values are not significant. (**E**) Calcium transients are shown as ΔF/F for spine (lighter color tone) and nearby dendritic shaft (darker color tone) evoked during local NMDA-spikes (blue) and spike timing-dependent plasticity (STDP) stimulation (one EPSP paired with three BAPs at 150 Hz; black). Inset: the dendritic shaft and spine corresponding to the traces shown in (**E**). line scan through spine and shaft (dashed green line, 2 ms per line). (**F**) Box plot showing EPSP amplitudes during control and post-NMDA-spike induction at 4 Hz. There was a significant change in the EPSP amplitude (148.48% ± 3.85% of control; p=0.002, n = 9). The gray dots represent the average EPSP for each cell, and the diamond represents the mean EPSP of the entire set. (**G**) Box plot showing EPSP amplitudes during control and post-pairing protocol (one EPSP with three BAPs at 150 Hz), delivered five times at 4 Hz. There was no significant change in EPSP amplitude (97.31% ± 3.64% of control; p=0.895, n = 4). The gray dots represent the average EPSP for each cell, and the diamond represents the mean EPSP of the entire set. (**H**) Box plot showing average ΔF/F evoked during local NMDA-spikes and during pairing protocol (one EPSP with three BAPs at 150 Hz) at proximal apical dendrites (127 ± 4.39 μm from soma). The calcium transients evoked during local NMDA-spikes were significantly higher than those evoked during pairing (p<0.0001). The gray dots represent the average ΔF/F for each cell, and the diamond represents the mean ΔF/F of the entire set. Dotted gray lines connect between pairs of control and post-induction values.

The online version of this article includes the following figure supplement(s) for figure 6:

**Source data 1.** Data for *Figure 6C, F-H*.

## Location-dependent plasticity in PCx pyramidal neurons

A key finding described in the literature with respect to the two main PCx pathways is their differential susceptibility to long-term plasticity. While associational IC inputs retain their capability for LTP changes throughout adulthood, LOT inputs were reported to have a critical period during development for long-term plasticity changes. Previous work has shown that NMDA-R-dependent LTP can be induced robustly in IC synapses both by burst stimulation and by STDP induction protocols (*Haberly et al., 1994*; *Johenning et al., 2009*; *Kanter et al., 1996*; *Neville and Haberly, 2004*). However, induction of LTP was less consistent with LOT inputs (*Haberly et al., 1994*; *Jung et al., 1990*; *Neville and Haberly, 2004*; *Roman et al., 1993*) and was either very small and inconsistent (10–15% potentiation) using theta burst protocols (*Franks and Isaacson, 2005*; *Jung et al., 1990*; *Kanter and Haberly, 1990*; *Poo and Isaacson, 2007*; *Roman et al., 1993*), or absent using STDP protocols (*Johenning et al., 2009*).

The failure to induce LTP with STDP protocols in distally located LOT synapses is in line with reports from other types of pyramidal neurons (*Gordon et al., 2006*; *Kampa et al., 2007*; *Larkum et al., 2009*; *Letzkus et al., 2006*; *Nevian et al., 2007*; *Sandler et al., 2016*; *Sjöström and Häusser, 2006*; *Sjöström et al., 2008*) and is consistent with the severe attenuation of BAPs to the distal portions of the apical tree (*Bathellier et al., 2009*; *Johenning et al., 2009*; *Kumar et al., 2018*) and with the high density of A-type potassium channels in the distal apical tree of PCx neurons (*Johenning et al., 2009*). Here, we confirm these findings and show that STDP protocols cannot induce LTP in the distally located LOT synapses but can reliably induce LTP in the more proximally located IC synapses in apical and distal basal trees.

In contrast to the prevailing view, here we show that the local NMDA-spike induction protocol was very efficient in generating robust and powerful LTP of LOT synapses. NMDA-spikes powerfully depolarize the distal locations of the apical tree and are associated with large calcium influx locally (*Kumar et al., 2018*; *Major et al., 2008*), and thus serve as a strong postsynaptic signal for induction of plasticity. Our results are consistent with a previous study in CA3 pyramidal neurons showing the necessity of local NMDA-spikes for plasticity in these neurons (*Brandalise et al., 2016*). Moreover, similar to our findings, previous reports show that dendritic spikes in general can serve as strong and efficient mediators for LTP in CA1 pyramidal neurons (*Remy and Spruston, 2007*). Importantly, our results are also consistent with in vivo data showing enhanced strength between the OB inputs and the PCx during olfactory learning in adulthood (*Chu et al., 2016*; *Cohen et al., 2008*; *Cohen et al., 2015*; *Roman et al., 1993*). We assume that the reason for the inability of prior studies to induce robust LTP of LOT inputs was related to the fact their induction protocols did not reliably generate local NMDA-spikes in distal apical dendrites (*Franks and Isaacson, 2005*; *Kanter and Haberly, 1993*; *Poo and Isaacson, 2007*). Our results thus are consistent with a mechanism where NMDA-spike-mediated potentiation of co-activated LOT synapses clustered on the same dendritic segment can be

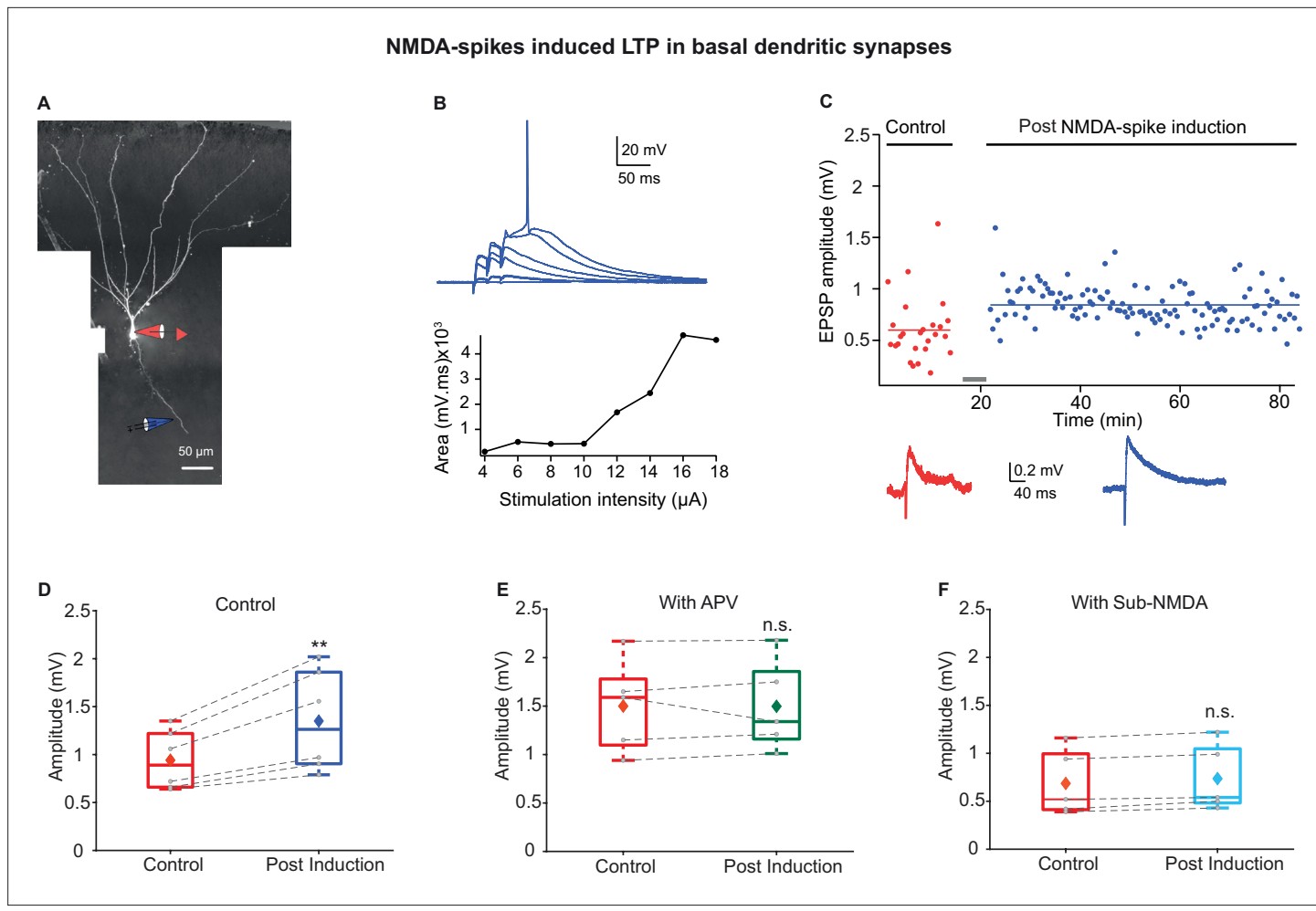

**Figure 7.** Long-term potentiation (LTP) of basal inputs induced by local NMDA-spikes in piriform cortex (PCx) pyramidal neurons. (**A**) Fluorescence image reconstruction of a layer IIa pyramidal neuron filled with CF633 (200 μM) and OGB-1 (200 μM) via a somatic patch electrode (red electrode). Focal stimulation was performed using a theta electrode placed nearby a distal basal dendrite (blue electrode; 144.83 ± 9.21 μm from soma), while recording at soma. (**B**) Voltage responses evoked by gradually increasing synaptic stimulation consisting of a burst of three pulses at 50 Hz. With gradually increasing stimulus intensity, an NMDA-spike was evoked (top). The peak voltage response is presented as a function of stimulation intensity for the voltage responses shown above (bottom). (**C**) Amplitude of single EPSPs is represented over time for control stimulation and NMDA-spike induction protocol at a basal dendrite. EPSPs were recorded at 0.033 Hz before and after the induction. Down: average EPSPs during control (red) and post-NMDA-spike induction paradigm (blue). (**D**) Box plot depicting EPSP amplitudes pre- and post-NMDA-spike induction protocol. NMDA-spike induction protocol induced potentiation of the control EPSP, 140.64% ± 4.5% of control (p=0.006526, n = 6). The gray dots represent the average EPSPs for each cell, and the diamond represents the mean EPSPs of the entire set. (**E**) Box plot showing EPSP amplitudes pre- and post-LTP induction protocol in the presence of APV. No significant change in EPSP amplitudes was observed, 100.69% ± 4.27% of control (p=0.9948, n = 5). (**F**) Box plot showing EPSP amplitudes pre- and post-LTP induction with sub-NMDA EPSPs at 4 Hz. No significant change in EPSP amplitudes was observed, 108.73% ± 3.13% of control (p=0.825, n = 5). In the box plots, the gray dots represent the average EPSPs of each experiment, and the diamond represents the mean of the entire set. Dotted gray lines connect between pairs of control and post-induction values. Data for (**B–F**) can be found in *Figure 7—source data 1*. See also *Figure 7—figure supplement 1*.

The online version of this article includes the following figure supplement(s) for figure 7:

**Source data 1.** Data for *Figure 7B-F*.

**Figure supplement 1.** Example experiments showing NMDA-spikes are necessary for long-term potentiation (LTP) of intracortical (IC) synapses in basal dendrites.

**Figure supplement 1—source data 1.** Data for *Figure 7—figure supplement 1B and D*.

used for storage of different olfactory combinations onto different dendritic branches (*Kumar et al., 2018*; *Poirazi et al., 2003*).

In proximal IC synapses, we find that despite the significantly larger calcium influx evoked by NMDA-spikes compared with STDP, the degree of potentiation of apical IC synapses was smaller with NMDA-spike protocol compared with STDP. This indicates that the amount of calcium entry per se is not the only variable determining the degree of potentiation. The kinetics and source of calcium entry may also influence potentiation (*Inglebert and Debanne, 2021*; *Zhou et al., 2005*), as well as possible involvement of neuromodulators. See, for example, *Gordon et al., 2006*, where we showed that in distal basal dendrites of layer 2–3 neocortical pyramidal neurons, brain-derived neurotrophic factor (BDNF) was a necessary requirement to gate plasticity in addition to calcium entry. All these factors may impact downstream transduction mechanisms that ultimately determine the degree of LTP. Further studies are needed to clarify this point.

## Learning mechanisms in piriform cortex

The anatomical arrangement of the OB LOT inputs is such that these inputs terminate broadly throughout the PCx and single pyramidal neurons receive inputs from multiple broadly distributed olfactory glomeruli (*Miyamichi et al., 2011*; *Nagayama et al., 2010*; *Sosulski et al., 2011*). This anatomical arrangement enables a large space of odor combinations to randomly terminate on different neurons and dendritic branches of PCx pyramidal neurons.

A leading theory of odor learning in PCx postulates that an odor representation is generated via strengthening of IC recurrent synapses activated during odor presentation, thus forming odor-specific ensembles in the PCx (*Haberly, 2001*; *Hasselmo and Barkai, 1995*; *Wilson and Sullivan, 2011*). The specificity to odor is enabled by strengthening the connectivity only between neurons that respond to the same set of LOT inputs. Upon exposure to the odor, these neurons will fire, and as a result the connectivity between these neurons will be strengthened by associative LTP mechanisms. Upon repeated exposures to the odor, a stronger and more robust activation of this odor-specific neuronal ensemble will take place due to the potentiated IC recurrent connections (*Haberly, 2001*). Experimental evidence supports the strengthening of IC inputs by an associative LTP mechanism, where IC and LOT inputs are co-activated (*Johenning et al., 2009*; *Kanter and Haberly, 1993*). In support of this idea, plasticity of IC synapses in vivo was shown in multiple studies (*Ghosh et al., 2015*; *Saar et al., 2001*; *Saar et al., 2002*; *Saar et al., 2012*).

Here, we propose an additional component to this model, where LOT inputs coding for a specific odor and terminating on the same distal dendritic branch could be self-potentiated via the NMDA-spike mechanism and a memory trace of the odor will be formed in PCx. Such a mechanism is consistent with the in vivo findings showing potentiation of bulb inputs to PCx during complex olfactory learning (*Cohen et al., 2008*; *Cohen et al., 2015*). NMDA-spike-based plasticity of LOT inputs can potentially contribute to odor representation in multiple ways: first, formation of an odor memory trace at the LOT with branch-specific dendritic plasticity mechanisms will increase the computational and storage capacity of PCx neurons (*Weber et al., 2016*; *Wu and Mel, 2009*). Instead of a cell assembly representing a single odor, it can be stored in single dendrites. Moreover, directly strengthening odor representations via such a branch-specific plasticity mechanisms of LOT synapses is expected to augment the single-neuron capacity to represent high-order combinations of odors (*Kumar et al., 2018*; *Wu and Mel, 2009*). Second, one important difference of NMDA-spike plasticity versus STDP can be the speed of acquiring and memorizing odors. Odor representation can be formed rapidly after only a few sniffs (*Rinberg et al., 2006*; *Uchida and Mainen, 2003*). NMDA-spike-mediated potentiation of LOT synapses can serve for such a rapid learning as it requires only a few spikes, while STDP mechanisms require many repetitions (see also *Lisman and Spruston, 2005*). Third, a plasticity rule that is not dependent on axo-somatic output action potentials remains isolated from the ongoing output activity of neurons, and thus is better protected from erosion and overwriting of memories that depend on somatic action potentials such as associative memories dependent on STDP mechanisms (*Bono and Clopath, 2017*). Thus, encoding of odor memories via local NMDA-spikes in distal dendritic compartments can contribute to more resilient and robust representation of odors. Finally, NMDA-spikes, although local, still contribute prolonged subthreshold voltage to the soma and other more proximal dendritic sites (*Kumar et al., 2018*). In this manner, NMDA-spikes can interact with other proximal inputs originating within the local network or other higher cortical areas to generate plasticity of coincident IC inputs using STDP mechanisms. Thus, this robustly potentiated odor information can later assist in generating cell assemblies representing associations between odors and

contextual information from local and from higher order regions such as orbitofrontal cortex and amygdala, which can potentially allow the assignment of cognitive and emotional value to odors. Upon reactivation of the same odor, these potentiated odor-specific LOT inputs could recall the odor more easily and further contribute to more efficient recruitment of odor-specific ensembles, ultimately enabling a more sensitive and reliable recall of odors (*Weber et al., 2016*; *Wu and Mel, 2009*).

### Feasibility of NMDA-spike plasticity mechanisms in vivo

To get more insight with regard to the potential feasibility of NMDA-spikes to occur in vivo, we estimated the probability of their occurrence under in vivo-like conditions based on the following assumptions from the literature: (1) typically layer IIb pyramidal neurons (PNs) in adult mice have 10 terminal apical branches (*Moreno-Velasquez et al., 2020*). (2) The size of the LOT band is ~100 µm (*Bekkers and Suzuki, 2013*; *Moreno-Velasquez et al., 2020*). (3) Typical number of spine density is at least 1 spine/ µm, which would result in at least ~100 spines per single-terminal branch at the LOT band. This band is almost exclusively innervated by LOT axons (*Bekkers and Suzuki, 2013*; *Giessel and Datta, 2014*; *Kumar et al., 2018*). Thus, per terminal branch there are ample of LOT synapses given that ~10 synapses are needed to initiate a local NMDA-spike (*Antic et al., 2010*; *Major et al., 2013*; *Moreno-Velasquez et al., 2020*; *Polsky et al., 2009*). However, a critical question relates to the statistics of LOT activation during a natural odor stimulation. *Srinivasan and Stevens, 2018* estimated that each piriform neuron randomly receives ~0.64 synapses from each single glomerulus. Assuming 110 glomeruli are activated by a typical odor yields activation of 70 LOT synapses per neuron (*Moreno-Velasquez et al., 2020*; *Srinivasan and Stevens, 2018*). Further, assuming random independent connectivity of LOT inputs to layer IIb PNs (*Giessel and Datta, 2014*; *Moreno-Velasquez et al., 2020*) and random distribution of LOT synapses over the terminal branches, we calculated the probability that at least one terminal dendritic branch will be simultaneously activated by 10 or more random LOT synapses (the estimated number for NMDA-spike initiation) in any given neuron to be 15% (*Figure 3—figure supplement 2*).

## Materials and methods

**Key resources table**

| Reagent type (species) or resource | Designation | Source or reference | Identifiers | Additional information |
|---|---|---|---|---|
| Genetic reagent (*Mus musculus*) | C57BL/6J | Jackson laboratory | RRID:IMSR_JAX:000664 | |
| Strain, strain background (*Rattus norvegicus*) | Wistar male/female | Envigo | RccHan:WIST | |
| Recombinant DNA reagent | pAAV.CAG. hChR2 (H134R)-mCherry.WPRE. SV40 | Addgene | ID: 100054 | |
| Chemical compound, drug | CF633 hydrazide | Biotium | Cat.#: 92156 | 200 µM |
| Chemical compound, drug | Oregon Green 488 BAPTA-1 Hexapotassium salt | Invitrogen | Cat.#: 06806 | 200 µM |
| Chemical compound, drug | MNI-Caged-L-glutamate | Tocris Bioscience | Cat.#: 1490/10 | 5–10 mM |
| Chemical compound, drug | D-AP5 | Tocris Bioscience | Cat.#: 0106/1 | 50 µM |
| Chemical compound, drug | Baclofen | Tocris Bioscience | Cat.#: 0796/10 | 100 µM |
| Software, algorithm | Igor Pro 8 | WaveMetrics | RRID:SCR_000325 | |

*Continued on next page*

*Continued*

| Reagent type (species) or resource | Designation | Source or reference | Identifiers | Additional information |
|---|---|---|---|---|
| Software, algorithm | pClamp 10 | Molecular devices | RRID:SCR_011323 | |
| Software, algorithm | Excel | Microsoft | RRID:SCR_016137 | |
| Software, algorithm | Adobe Illustrator | Adobe | RRID:SCR_010279 | |
| software, algorithm | MATLAB | MathWorks | RRID:SCR_001622 | |

## Electrophysiology and calcium imaging

All animal procedures were done in accordance with the guidelines established by the NIH on the care and use of animals in research and were confirmed by the Technion Institutional Animal Care and Use Committee.

Acute coronal brain slices 300 μm thick were prepared from the anterior PCx of Wistar rats (male and female) 4–6 weeks old or mice 7–12 weeks old. The entire brain was removed and placed in ice-cold sucrose solution, maintained under 5°C temperature and saturated with 95% oxygen and 5% $CO_2$. The sucrose solution contained (in mM) 2.5 KCl, 1.25 $NaH_2PO_4$, 25 $NaHCO_3$, 7 $MgCl_2$, 7 dextrose, 9 ascorbic acid, and 300 sucrose. The slices were kept in an artificial cerebrospinal fluid (ACSF) at 34–36°C for 30 min recovery and later kept at room temperature. The ACSF solution contained (in mM) 125 NaCl, 25 glucose, 3 KCl, 25 $NaNCO_3$, 2 $CaCl_2$, 1.25 $NaH_2PO_4$, 1 $MgCl_2$, pH 7.4.

During experiment, cells were visualized with a confocal scanning microscope equipped with IR illumination and Dodt gradient contrast video microscopy. Whole-cell patch-clamp recordings were performed on visually identified layer II pyramidal neurons using an Axon amplifier (multi-clamp 700 A). For patching, glass electrodes (6–9 MΩ) were prepared from thick-walled (0.2 mm) borosilicate glass capillaries (2 mm) using a micropipette puller (P-97, Sutter Instrument). The intracellular pipette solution contained (in mM) 135 $K^+$-gluconate, 4 KCl, 4 Mg-ATP, 10 $Na_2$-phosphocreatine, 0.3 Na-GTP, 10 HEPES, 0.2 OGB-1, 0.2 CF 633, pH 7.2.

Fluorescence confocal microscopy was performed on an upright BX61WI Olympus microscope equipped with 60× (Olympus 0.9 NA) water-immersion objective. Neurons were filled with calcium-sensitive dye OGB-1 (200 μM; Invitrogen) and CF 633 (200 μM; Biotium) to visualize and collect calcium signals during focal stimulation from the apical and basal dendritic trees. Calcium transients were recorded in a line-scan mode at 500 Hz. The temperature of the slice bath was maintained at 34°C for the entire duration of the experiment.

## Focal synaptic stimulation

Focal synaptic stimulation at apical and basal dendrites of pyramidal neurons was performed using theta-glass pipette (borosilicate; Hilgenberg), placed in close proximity (5–10 μm) to the desired dendritic segment guided by the fluorescent image of the dendrite and Dodt contrast image of the slice. The theta-stimulation electrodes were filled with CF 633 (0.1 mM; Biotium) diluted with filtered ACSF. Current was delivered through the electrode (short bursts of three pulses at 50 Hz) at varying intensities using a stimulus isolator (ISO-Flex; AMPI). The efficacy and location of the stimulation was verified by simultaneous calcium imaging evoked by small EPSPs and their localization to a small segment of the stimulated dendrite.

## Glutamate uncaging

Caged MNI-L-glutamate (Tocris, UK) was delivered locally nearby a dendritic segment in LOT region using pressure ejection (5–10 mbar) from a glass electrode (2–3 μm in diameter) containing 5–10 mM caged glutamate. The electrode was placed 20–30 μm from the dendrite of interest, and the caged glutamate was photolyzed using a 1 ms laser pulse (351 nm, Excelsior, Spectra Physics) using point scan mode (Olympus FV1000). Laser power at the objective with this scanning mode was 1–10 μW.

## Drug application

All experiments were performed in the presence of one of the two anion gamma-aminobutyric acid (GABA$_A$) blockers, bicuculline (1 μM; Sigma) or gabazine (10 μM; Tocris Bioscience). There were no differences in outcomes observed between these two GABA$_A$ blockers. In some experiments, the NMDA-R blocker (APV 50 μM; Tocris Bioscience) was added to the ACSF perfusion solution 20–30 min before the start of the recording. GABA$_B$ agonist baclofen (100 μM) was added in some experiments to selectively silence IC connections.

## Stereotaxic virus injections

C57B mice 4–7 weeks old were anesthetized by isoflurane inhalation (4% for induction and 1.5–2% during surgery) and mounted on a stereotaxic frame. Ketoprofen (5 mg/kg) and buprenorphine (0.1 mg/kg) were administered. Labeling of the LOT fibers was performed by unilateral injection of pAAV.CAG.hChR2(H134R)-mCherry.WPRE.SV40 (Addgene) into the OB at coordinates +4.0 mm and +4.5 mm AP from Bregma, + 1.0 mm ML at depth 1, 1.5 and 2 mm (20 nL in each depth). The injections were made through the thinned skull using a hydraulic micromanipulator (M0-10 Narishige). Following the surgery, ketoprofen and buprenorphine were administered to the animal for two consecutive days. The animals were then returned to their home cage for a period of minimum 3 weeks to ensure full recovery from the surgery and expression of the injected virus.

## Long-term plasticity induction protocols

Control EPSPs were acquired at 0.033 Hz before (10–15 min) and after the induction protocol. Stability of the recording was ensured by monitoring the resting membrane potential, and only experiments in which the membrane potential was not changed more than 3 mV were included. STDP was induced by pairing a single EPSP, followed 8 ms later with a burst of postsynaptic BAPs (three BAPs at 150 Hz) evoked by somatic current injection. Triplets of this EPSP-BAPs pairing (at 20 Hz) were repeated 40 times at 5 s interval. In some experiments, postsynaptic BAPs were generated by synaptic stimulation of proximal apical dendrites (~100 μm from soma).

NMDA-spike LTP protocol consisted of 2–23 NMDA-spikes (evoked by a burst of three stimuli at 50 Hz) repeated at 4 Hz, the exploratory sniff frequency in rats. Same protocol was repeated with synaptic stimulation, which was subthreshold for NMDA-spike initiation (sub-NMDA EPSPs) and in the presence of APV (50 μM).

For optogenetic LTP induction protocol, stimulation was performed with a Mightex Polygon micro-mirror device using a 470 nm blue LED light source. Light was delivered to the top of the slice through ×60 objective (Olympus 0.9 NA). Opto-EPSPs were evoked by 5 ms light pulses delivered every 30 s. To evoke an NMDA-spike by optogenetically activating LOT fibers, three light pulses of 5 ms each were delivered at 50 Hz.

## Data analysis and statistical procedure

The sample size was chosen based on standards used in the field with similar experimental paradigms. Importantly most of our experiments involve examining a variable on the same neuron, and thus the sources of variability are smaller in these type of experiments.

All electrophysiological data collected were analyzed using Clampfit 10.3 (Axon instruments), Igor Pro software (5.01, Wave metrics), Microsoft Office 365, and an in-house MATLAB software. All data are presented as mean ± SEM. Two-tailed paired Student's t-test was used for testing statistical significance of the data. All neurons in which the recordings deteriorated, as measured by the series resistance of the electrode, were excluded from the averages.

## Acknowledgements

We thank Y Schiller for helpful discussions throughout the project and helpful comments on the manuscript and B Mel for helpful comments on the manuscript. We thank Irena Reiter for excellent technical assistance. This study was supported by Israeli Science Foundation (JS), German Israeli Foundation (JS), and Prince funds (JS).

# Additional information

### Funding

| Funder | Grant reference number | Author |
| --- | --- | --- |
| Israel Science Foundation | | Jackie Schiller |
| Prince Center for Neurodegenerative Diseases | | Jackie Schiller |

The funders had no role in study design, data collection and interpretation, or the decision to submit the work for publication.

### Author contributions

Amit Kumar, Data curation, Formal analysis, Methodology, Visualization, Writing - original draft; Edi Barkai, Conceptualization, Writing - original draft; Jackie Schiller, Conceptualization, Data curation, Formal analysis, Funding acquisition, Methodology, Supervision, Writing - original draft, Writing - review and editing

### Author ORCIDs

Amit Kumar  http://orcid.org/0000-0003-0674-3641
Edi Barkai  http://orcid.org/0000-0002-7325-4269
Jackie Schiller  http://orcid.org/0000-0001-9182-7166

### Ethics

All animal procedures were done in accordance with guidelines established by NIH on the care and use of animals in research and were confirmed by the Technion Institutional Animal Care and Use Committee (Permit number IL-012-01-18).

### Decision letter and Author response

Decision letter https://doi.org/10.7554/eLife.70383.sa1
Author response https://doi.org/10.7554/eLife.70383.sa2

# Additional files

### Supplementary files

• Transparent reporting form

### Data availability

The data generated or analysed during this study are included in the manuscript and Supporting files for Figure 1C-G, Figure 2B,D, Figure 3B,D, F-H, Figure 4 B,D, Figure 5B-D, Figure 6 C-D and Figure 7 B-F, are loaded.

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
