## [Editor Report]

This article investigates the plastic properties of synapses impinging on pyramidal neurons in the piriform cortex from the lateral olfactory tract (LOT) and intracortical inputs. These findings uncover some of the location and pathway-dependent plasticity rules and challenge the notion that LOT inputs (carrying direct odor information from the bulb) become ‘hardwired’ after the critical period. The results provide novel information about how activity and experience alter synaptic communication in the olfactory circuit in a synapse-type specific manner.

---

## [Decision Letter]

**Decision letter after peer review:**

Thank you for submitting your article "Plasticity of olfactory bulb inputs mediated by dendritic NMDA-spikes in piriform cortex" for consideration by *eLife*. Your article has been reviewed by 3 peer reviewers, and the evaluation has been overseen by a Reviewing Editor and John Huguenard as the Senior Editor. The following individuals involved in review of your submission have agreed to reveal their identity: Sonia Gasparini (Reviewer #1); Ian G Davison (Reviewer #2).

Essential revisions:

1. The authors would need to identify the mechanisms of why distally located LOT synapses can trigger strong LTP via NMDA-spikes, but those more proximally located IC synapses cannot. The authors should explore this question in more detail experimentally.

2. The manuscript should have an extended discussion on the type of input required to initiate dendritic NMDA spikes and the likelihood of this phenomenon happening in vivo. Connectivity data could help to make rough estimates, and would improve the manuscript. quite significantly.

*Reviewer #1 (Recommendations for the authors):*

My suggestions are mainly to make some aspects of the manuscript clearer.

Since the EPSP amplitudes are plotted as paired sampled measurements from before and after the induction protocol (whether it is pairing or NMDA spikes), it would be more informative to connect the pre- and post- data from each cell with a line.

For Figure 5, it is said that dendritic spikes were obtained either with glutamate uncaging or more intense optogenetic activation of LOT input. Were the data for the two methods pooled together? If so, maybe the authors could use different colors to distinguish the two induction methods when connecting the pre- and post-data as described in the comment above.

The authors mention OGB-1 and details of Ca^2+^ imaging in the methods, but there don't seem to be data of Ca^2+^ transients reported in the results. This detail could be distracting for readers.

Line 23-25 this sentence should be followed by a reference

Lines 148-156 the manuscript would flow more easily if the description of figure 3F-H was placed after line 133

*Reviewer #2 (Recommendations for the authors):*

Some more detail addressing why the two cellular locations have such different plasticity rules would be useful. Backpropagation presumably accounts for the weak distal STDP effects, but it's not clear why there is no NMDA-spike plasticity more proximally. Do NMDA spikes cause equivalent nonlinearities at both sites? Could lack of plasticity be due to weaker NMDA spikes at proximal synapses? The single example shown in 6B looks less pronounced than in 3B / 5B. A more thorough quantification of the NMDA spike properties or nonlinear input-output function at the different sites could help address this.

Even if speculative it could be useful to add some more discussion on how LOT plasticity could be useful for odor processing. How could these different plasticity rules contribute to responses in piriform beyond responding to clustered inputs? Would this increase the specificity of the piriform cells that first fire in response to feedforward distal LOT input?

---

## [Author Response]

Essential revisions:1. The authors would need to identify the mechanisms of why distally located LOT synapses can trigger strong LTP via NMDA-spikes, but those more proximally located IC synapses cannot. The authors should explore this question in more detail experimentally.

Following the reviewers comments we have performed additional experiments to clarify this point. As noted by reviewer # 2, for these proximally located IC synapses we used smaller NMDA-spikes to avoid back-propagating action potentials. However, we realize that by doing so we were not able to explore in full the ability of full-blown NMDA-spikes to induce plasticity at these synapses. We have performed new experiments where we enabled full blown NMDA- spikes and controlled for the possible contribution of back-propagating action potentials. In addition, we have also performed calcium imaging to compare calcium entry with NMDA-spikes compared to STDP protocol. Using full blown NMDA-spikes we were able to induce LTP in proximal IC synapses but to a smaller extent compared to STDP in same locations despite larger calcium entry with NMDA-spikes.

These results are now described in a new Figure 6.

2. The manuscript should have an extended discussion on the type of input required to initiate dendritic NMDA spikes and the likelihood of this phenomenon happening in vivo. Connectivity data could help to make rough estimates, and would improve the manuscript. quite significantly.

Following the reviewers comment we have added an extended discussion on page 12 (in non-tracked version of the manuscript) along with a new Figure 3—figure supplement 2 where we included our estimations based on the data from the literature.

Reviewer #1 (Recommendations for the authors):My suggestions are mainly to make some aspects of the manuscript clearer.Since the EPSP amplitudes are plotted as paired sampled measurements from before and after the induction protocol (whether it is pairing or NMDA spikes), it would be more informative to connect the pre- and post- data from each cell with a line.

Following the reviewer comment, we added this information to the box-plots and exchanged respectively all the appropriate panels in the figures (Figures 1 to 7 and Figure 1—figure supplement 1).

For Figure 5, it is said that dendritic spikes were obtained either with glutamate uncaging or more intense optogenetic activation of LOT input. Were the data for the two methods pooled together? If so, maybe the authors could use different colors to distinguish the two induction methods when connecting the pre- and post-data as described in the comment above.

We added this information by indicating the experiments in the box plot by color (Figure 5D).

The authors mention OGB-1 and details of Ca^2+^ imaging in the methods, but there don't seem to be data of Ca^2+^ transients reported in the results. This detail could be distracting for readers.

The reviewer is right. We actually performed regularly calcium imaging to check the focality of our synaptic stimulation but this is not included in the manuscript rather served as our own internal check. However, now following the reviewers comments we added calcium imaging data describing the comparison between calcium transients evoked by NMDA-spikes compared to STDP in proximal locations. This is described in Figure 6E, H.

Line 23-25 this sentence should be followed by a reference

Added (Wilson and Sullivan 2011).

Lines 148-156 the manuscript would flow more easily if the description of figure 3F-H was placed after line 133

We thank the reviewer for this comment. We changed the text accordingly.

Reviewer #2 (Recommendations for the authors):Some more detail addressing why the two cellular locations have such different plasticity rules would be useful. Backpropagation presumably accounts for the weak distal STDP effects, but it's not clear why there is no NMDA-spike plasticity more proximally. Do NMDA spikes cause equivalent nonlinearities at both sites? Could lack of plasticity be due to weaker NMDA spikes at proximal synapses? The single example shown in 6B looks less pronounced than in 3B / 5B. A more thorough quantification of the NMDA spike properties or nonlinear input-output function at the different sites could help address this.

We thank the reviewer for this comment, following which we performed additional experiments and reanalyzed the data. As suggested by the reviewer, NMDA-spikes in the proximal apical dendrites that served for the induction protocol of the original version of the manuscript were indeed smaller than the full-blown NMDA spikes that can be generated in proximal location (average peak amplitudes of 32.7±4.6 mV and 47.4±5.8 mV and average area under curve of 3725±461 mV*ms and 7741±974 mV*ms for small and full blown NMDA-spikes respectively; see also Kumar et al. 2018). The reason we used smaller NMDA-spikes was to avoid initiation of BAPs during the induction period. However, following the reviewer’s comment to clarify this point we did further experiments: 1. We examined the ability of full blown proximal NMDA-spikes to induce plasticity in IC synapses. A similar NMDA-spike induction protocol (4-10 NMDA-spikes at 4Hz) also induced potentiation of proximal IC synapses (127 ± 4.39 microns from soma) (Figure 6A-D), but to a smaller extent compared to distal LOT synapses and even smaller than potentiation induced by STDP protocol in these synapses. Post induction, the proximal IC EPSP amplitude was 148.48 ± 4.1% of the control (Figure 6F; p = 0.00204; n = 9; p=0.0013 for comparison of proximal versus distal NMDA-spike potentiation; p=0.0127 for comparison with STDP in IC synapses). The number of NMDA spikes needed for this potentiation was between 4-10 NMDA spike repetitions. 2. To control for the contribution of these BAPs, we repeated the induction protocol but instead of using NMDA-spikes we used pairing BAPs and local EPSPs for 5 repetitions (1 EPSP paired with 3BAPs at 150 Hz repeated 5 times at 4 Hz). In this case we did not observe potentiation of these proximal IC synapses (Figure 6G), thus we concluded NMDA-spikes were crucial for the potentiation of IC synapses with the NMDA-spike protocol. 3. We measured the local calcium transients in active spines and neighboring shafts following STDP protocol activation (pairing BAPS and EPSPs) compared to local NMDA spikes in proximal IC. Interestingly we find, calcium transients evoked by NMDA-spikes both in shafts and spines, were significantly larger than those evoked by STDP stimulation (Figure 6Eand 6H; p<0.0001) despite the degree of potentiation with STDP protocol was higher (p=0.0127) compared to the NMDA-spike protocol. This results indicate the amount of calcium entry per se is not the only variable determining the degree of potentiation. See for example Gordon et al. 2006 where we showed that in distal basal dendrites of layer 2-3 neocortical neurons BDNF was a necessary requirement to gate plasticity in addition to calcium entry.

These new experiments were added to the revised version and are replacing the previous experiments (page 7 lines 15<milestone-start />3<milestone-end />-1<milestone-start />68<milestone-end /> in non-tracked version of the manuscript and new Figure 6).

Even if speculative it could be useful to add some more discussion on how LOT plasticity could be useful for odor processing. How could these different plasticity rules contribute to responses in piriform beyond responding to clustered inputs? Would this increase the specificity of the piriform cells that first fire in response to feedforward distal LOT input?

Following the reviewer’s comment, we elaborated the discussion on the possible contribution of NMDA spikes plasticity to odor representation. We will speculate about how NMDA spike evoked LTP could uniquely contribute to plasticity and learning of odor in PCx: First, formation of an odor memory trace at the LOT with branch specific dendritic plasticity mechanisms, will increase the computational and storage capacity of PCx neurons (Weber et al. 2016; Wu and Mel. 2009). Instead of a cell assembly representing a single odor, it can be stored in single dendrites. Moreover, directly strengthening odor representations via such a branch specific plasticity mechanisms of LOT synapses, is expected to augment the single neuron capacity to represent high-order combinations of odors (Kumar et al. 2018; Wu and Mel. 2009). Second, one important difference of NMDA-spike plasticity versus STDP can be the speed of acquiring and memorizing odors. Odor representation can be formed rapidly after only few sniffs (Rinberg et al. 2006; Uchida and Mainen. 2003). NMDA-spike mediated potentiation of LOT synapses can serve for such a rapid learning as it requires only few spikes, while STDP mechanisms require many repetitions. (see also Lisman and Spruston. 2005). Third, a plasticity rule which is not dependent on axo-somatic output action potentials remains isolated from the ongoing output activity of neurons, and thus is better protected from erosion and overwriting of memories which depend on somatic action potentials such as associative memories dependent on STDP mechanisms (Bono and Clopath. 2017). Thus, encoding of odor memories via local NMDA-spikes in distal dendritic compartments can contribute to more resilient and robust representation of odors. Finally, NMDA-spikes although local, still contribute prolonged subthreshold voltage to the soma and other more proximal dendritic sites (Kumar et al. 2018). In this manner NMDA-spikes can interact with other proximal inputs originating within the local network or other higher cortical areas to generate plasticity of coincident IC inputs using STDP mechanisms. Thus, this robustly potentiated odor information, can later assist in generating cell assemblies representing associations between odors and contextual information from local and from higher order regions such as orbitofrontal cortex and amygdala, which can potentially allow the assignment of cognitive and emotional value to odors. Upon reactivation of the same odor, these potentiated odor specific-LOT inputs could recall the odor more easily and further contribute to more efficient recruitment of odor-specific ensembles, ultimately enabling a more sensitive and reliable recall of odors (Weber et al. 2016; Wu and Mel. 2009).

Following the reviewers’ comment in the revised manuscript we extended our discussion regarding the potential usefulness of NMDA spike mediated LOT plasticity in odor representation in the Discussion section (page 11-12 lines 26<milestone-start />9<milestone-end />-29<milestone-start />9<milestone-end /> in non-tracked version of the manuscript)